# Task Assignment of UAV Swarms Based on Deep Reinforcement Learning

Bo Liu [†][ID], Shulei Wang [†], Qinghua Li, Xinyang Zhao, Yunqing Pan and Changhong Wang *

Space Control and Inertial Technology Research Center, Harbin Institute of Technology, Harbin 150000, China; 17b904042@stu.hit.edu.cn (B.L.)
* Correspondence: cwang@hit.edu.cn
† These authors contributed equally to this work.

**Abstract:** UAV swarm applications are critical for the future, and their mission-planning and decision-making capabilities have a direct impact on their performance. However, creating a dynamic and scalable assignment algorithm that can be applied to various groups and tasks is a significant challenge. To address this issue, we propose the Extensible Multi-Agent Deep Deterministic Policy Gradient (Ex-MADDPG) algorithm, which builds on the MADDPG framework. The Ex-MADDPG algorithm improves the robustness and scalability of the assignment algorithm by incorporating local communication, mean simulation observation, a synchronous parameter-training mechanism, and a scalable multiple-decision mechanism. Our approach has been validated for effectiveness and scalability through both simulation experiments in the Multi-Agent Particle Environment (MPE) and a real-world experiment. Overall, our results demonstrate that the Ex-MADDPG algorithm is effective in handling various groups and tasks and can scale well as the swarm size increases. Therefore, our algorithm holds great promise for mission planning and decision-making in UAV swarm applications.

**Keywords:** UAV swarm; task assignment; deep reinforcement learning; Ex-MADDPG





## 1. Introduction

With their advantages of high altitude, low price, and strong substitutability, unmanned aerial vehicle (UAV) swarms are becoming increasingly prevalent in daily life. UAV swarm refers to a large number of UAVs with weak autonomous capability that can effectively perform complex tasks such as multi-aircraft formation and cooperative attack through information interaction and autonomous decision-making.

UAV swarm target-attacking is a complex process, including autonomous path planning, target detection, and task assignment, and it is almost impossible to design one algorithm to complete the whole combat process mentioned above. Therefore, this paper simplifies the whole UAV swarm target-attacking process into two parts: target detection and target assignment. The target-detection and target-assignment abilities of the UAV swarm affect the quality of mission accomplishment and are the most important parts of the swarm target-attacking system. However, different tasks have significant differences in operational objectives, time constraints, mission requirements, and other aspects. Simultaneously, sub-task coupling, self-organizing, and the large-scale nature of swarms pose great challenges for the mission planning and decision-making of the UAV swarm.

In recent years, the great potential of reinforcement learning (RL) within the swarm intelligence domain makes it an important approach to studying UAV swarm task assignment. However, RL task-assignment algorithms applied to UAV swarms still face a series of technical bottlenecks such as low sampling efficiency, difficult reward function design, poor stability, and poor scalability, so it is especially critical for scalable and robust task planning and decision-making algorithms to be designed for UAV swarms. Therefore, we

propose a scalable task-assignment method to deal with the dynamic UAV swarm task planning and decision-making problem in this paper.

### 1.1. Related Works

The UAV swarm task planning problem can be formulated as a complex combinatorial optimization problem [1] considering time constraints, task decomposition, and dynamic reallocation, which make it an NP-hard problem. The algorithms for task assignment are generally divided into optimization algorithms, heuristic algorithms, swarm intelligence algorithms, contract network, auction algorithms, and reinforcement learning algorithms.

The optimization algorithm aims to obtain the optimal solution according to the objective function under constraint conditions. Common optimization methods include enumeration algorithms, dynamic programming algorithms [2], integer programming algorithms [3], etc. The enumeration algorithm is the simplest task assignment algorithm and can only be used to solve problems of small size and low complexity. The dynamic programming algorithm is a bottom-up algorithm that establishes several sub-problems from the bottom and solves the whole problem by solving the sub-problems. The integer programming algorithm is the general name of a set of algorithms for solving integer programming problems, and it includes the Hungarian algorithm [4], the branch and bound method, etc.

The heuristic algorithm is an algorithm based on intuition or experience that aims to find feasible solutions to complex problems in a limited time. Common heuristic algorithms include the genetic algorithm [5] (GA), tabu search, simulated annealing [6] (SA), etc. Take GA as an example. GA was proposed by John Holland of the United States in the 1970s. The algorithm simulates genetic evolution in nature to search for the optimal solution. Wu et al. [7] combined the optimization idea of SA to improve the global optimization effect and convergence speed of GA. Martin et al. [8] dynamically adjusted the parameters of the genetic algorithm according to the available computational capacity, thus realizing the trade-off between computation time and accuracy.

The swarm intelligence algorithm is rooted in the concept of swarm intelligence, which is observed in nature. This algorithm addresses the task-assignment problem by exploring all feasible solutions in the problem space, including popular techniques such as Ant Colony Optimization Algorithm (ACO), Particle Swarm Optimization Algorithm (PSO), Grey Wolf (GW), etc. ACO mimics the foraging behavior of ants to determine an optimal solution [9]. Gao et al. [10] introduced a negative feedback mechanism to hasten the convergence of ACO, which has proved to be advantageous in solving large-scale task-assignment problems. Du et al. [11] devised a role-based approach for the ant colony to prioritize different search strategies, thus enhancing the efficiency of finding the global optimal solution. PSO, a random search algorithm that emulates bird feeding behavior, has been employed to tackle the task assignment problem [12,13]. Chen et al. [14] proposed a guided PSO approach that has been demonstrated to yield optimal task-assignment schemes, thereby improving the cooperative combat capability of multiple UAVs.

The contract network algorithm has been proposed to solve the workload balancing problem among unmanned aerial vehicles (UAVs) through a "bidding winning" mechanism [15]. Chen [16] presented a distributed contract network-based task assignment method to solve the communication-delay-constrained task assignment problem in multiple UAVs. The auction algorithm [17] mimics the human auction process to optimize the benefits of the UAV swarm system [18]. Liao [19] proposed a dynamic target-assignment algorithm using multi-agent distributed collaborative auctioning. Li et al. [20] introduced a result-updating mechanism where new and old tasks are reauctioned together, resulting in the most beneficial replanning results while satisfying real-time requirements. The effectiveness of this algorithm was demonstrated by the experimental results.

In recent years, deep RL (DRL), which combines RL and deep learning (DL), has emerged as an important research area in UAV control and decision-making. DRL alleviates the dimension explosion problem that easily occurs in traditional RL and has made great

breakthroughs in areas such as robot control [21–23], scheduling optimization [24,25], and multi-agent collaboration [26–29]. Ma [30] proposed a task-assignment algorithm based on Deep Q Network (DQN) to support UAV swarm operations, which significantly improves the success rate of UAV swarm combat. Huang [31] combines DQN with an evolutionary algorithm to optimize task-assignment results of traditional algorithms and can obtain assignment results dynamically.

In short, heuristic and swarm intelligence algorithms solve problems faster but produce suboptimal solutions that have poor scalability and flexibility. The improved distributed algorithms can only handle specific problems. Contract network and auction algorithms have high robustness and scalability, but both rely heavily on communication and computing capacities with poor flexibility. As for DRL, as the group size increases, there are problems such as spatial linkage of action states, dimensional explosion, and difficulty in determining the reward function.

*1.2. Contribution*

According to the aforementioned investigations, this paper presents Ex-MADDPG to solve dynamic task assignment problems for UAV swarms. The main distinguishing advantages of our algorithm are easy training, good scalability, excellent assignment performance, and real-time decision-making. We summarize the main contributions as follows.

(1)　We construct an extensible framework with local communication, a mean simulation observation model, and a synchronization parameter training mechanism to meet the scalability capability so that the strategies from small-scale system training can be directly applied to large-scale swarms.

(2)　Due to the poor assignment performance of the traditional DRL algorithm with increasing system scale, a multiple-decision mechanism is proposed to ensure the assignment performance of a large UAV swarm to perform complex and diverse tasks.

(3)　The practicality and effectiveness of the proposed Ex-MADDPG algorithm have been verified through simulation experiments carried out on the MPE simulation platform and a real-world experiment with nine drones and three targets. The results demonstrate that the proposed algorithm outperforms traditional task-assignment algorithms in various performance indicators, including task completion rate, task loss, number of communications, and decision time.

The paper is structured as follows. First, we describe the basic theory and background of DRL in Section 2. Then, the proposed method for the dynamic extensible task assignment problem is detailed in Section 3. Afterward, the efficiency of the proposed algorithm is verified in Sections 4 and 5. Finally, we draw the conclusion and outline possible future directions in Section 6.

## 2. Deep Reinforcement Learning Background

RL is a machine learning method for learning "what to do" to maximize utility. The agent must learn an optimal policy in the current situation through trial and error. Most RL algorithms are based on Markov Decision Processes (MDPs) for theoretical modeling, derivation, and demonstration. Figure 1 shows the interaction process between the agent and the environment in MDPs.

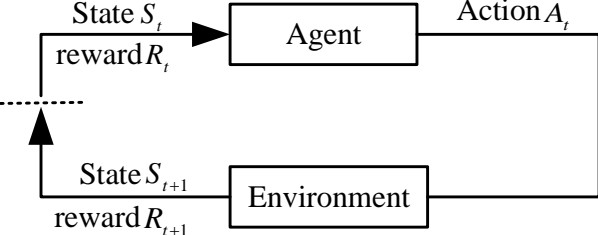

**Figure 1.** Interaction Process between Agent and Environment.

In MDPs, the decision-making process is described as the following quads:

$$(S, A, P, R) \tag{1}$$

where $S$ is the state set, $A$ is the action set, $S \times S \times A \to [0, 1]$, $R$ is the expected rewards of state–action, $S \times A \to R$, and $P$ is the state transition function.

$$
\begin{aligned}
p(s' \mid s, a) &= \Pr\{S_t \mid S_{t-1}, A_{t-1}\} \\
&= \sum_{r \in R} P(s', r \mid s, a)
\end{aligned}
\tag{2}
$$

In a UAV swarm system, a UAV observes a certain state $S_t \in S$ of the environment, then chooses an action $A_t \in A$ according to that state. In the next moment, according to the different selected action, the UAV will obtain a reward $R_{t+1} \in R$ from the state transfer function $P$ and enters a new state $S_{t+1}$. This process can be repeated to obtain the following Markov decision sequence:

$$S_0, A_0, R_1, S_1, A_1, R_2, S_2, A_2, S_3, \cdots \tag{3}$$

In 1989, Watkins [32] proposed the Q-learning algorithm by combining the time series difference learning method and optimal control, which is a great milestone for RL. In 2013, Mnih [33] proposed DQN by combining RL and DL and achieved the top level of human performance in a classic Atari game. In the DRL algorithm, Lillicrap [34] proposed a new algorithm that combines the DQN and Policy Gradient (PG) algorithm named Deep Deterministic Policy Gradient (DDPG) to solve the control problem in continuous action space effectively. The framework of DDPG is shown in Figure 2. DDPG samples the distribution of actions by improving the policy $\mu$ to obtain the specific action $A$. At this point, the reward function $R(s, a)$ is determined. The deterministic policy is $\mu_\theta : S \to A$, and its maximum objective function is:

$$J(\theta) = \mathbb{E}_{s \sim p^\mu}[R(s, a)] \tag{4}$$

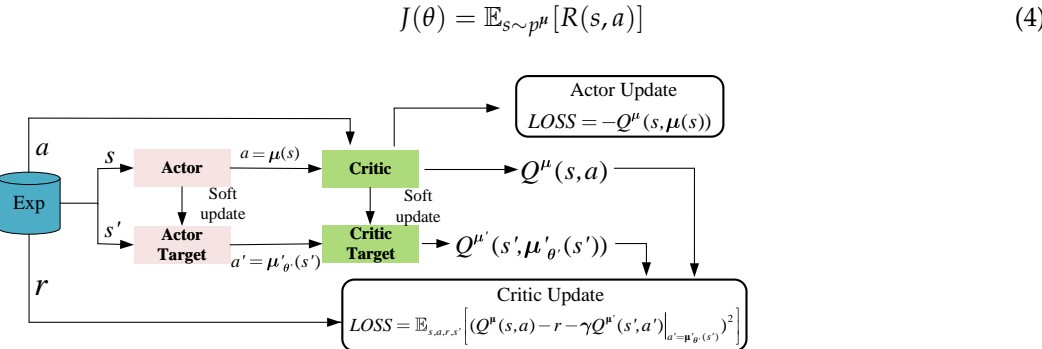

**Figure 2.** DDPG Algorithm Framework.

The corresponding gradient is

$$\nabla_\theta J(\theta) = \mathbb{E}_{s \sim D}\left[ \nabla_\theta \mu_\theta(a \mid s) \nabla_a Q^\mu(s, a)\big|_{a = \mu_\theta(s)} \right] \tag{5}$$

Equation (5) depends on $\nabla_a Q^\mu(s, a)$, and the action space of the DDPG algorithm must be continuous.

The critic network is updated as follows:

$$
\begin{aligned}
L(\theta) &= \mathbb{E}_{s,a,r,s'}\left[ (Q^\mu(s, a) - y)^2 \right] \\
y &= r + \gamma Q^{\mu'}(s', a')\big|_{a' = \mu'_{\theta'}(s')}
\end{aligned}
\tag{6}
$$

where $\mu$ is the actor prediction network and $\mu'$ is the target network. The gradient of the objective function of the actor network is:

$$\nabla_\theta J(\boldsymbol{\mu}) = \mathbb{E}_{s,a\sim D}\left[\nabla_\theta \boldsymbol{\mu}(a|s)\nabla_a Q^{\boldsymbol{\mu}}(s,a)|_{a=\boldsymbol{\mu}(s)}\right] \tag{7}$$

The loss function is:

$$L(\mu) = -Q^{\mu}(s,a) \tag{8}$$

Unlike the situation of single-agent training, the change in environment is not only related to the actions of the agent itself, but also related to the actions of other agents and the interaction between agents, which leads to the instability of multi-agent training. Based on the DDPG algorithm, Lowe [35] extended it to the case of multiple agents and proposed the MADDPG algorithm. It adopted a centralized training and distributed execution framework, as shown in Figure 3. It uses global information to guide the training of multi-agent, while in the process of execution, the agent will only make decisions based on its own observations. It greatly improves the convergence speed and training effect of the algorithm.

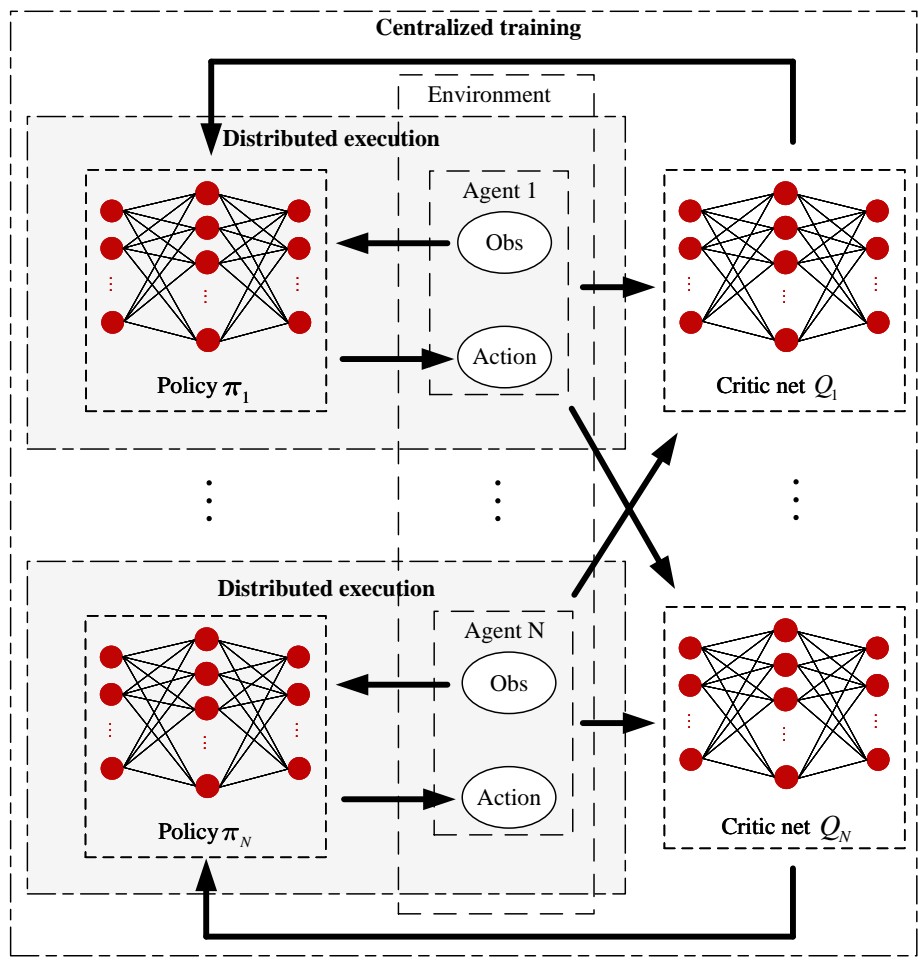

**Figure 3.** Centralized Training Distributed Execution Framework.

Considering $N$ agents, each agent's policy parameters can be written in the following form:

$$\boldsymbol{\theta} = \{\theta_1, \theta_2, \theta_3, \cdots, \theta_N\} \tag{9}$$

Its policy is

$$\boldsymbol{\pi} = \{\pi_1, \pi_2, \pi_3, \ldots, \pi_N\} \tag{10}$$

According to the PG algorithm, the gradient of the expected return $J(\theta_i) = \mathbb{E}[R_i]$ for agent $i$ can be obtained:

$$
\begin{aligned}
\nabla_{\theta_i} J(\theta_i) &= \mathbb{E}_{s \sim p^\mu, a_i \sim \pi_i}[\mathcal{L}(\theta_i)] \\
\mathcal{L}(\theta_i) &= \nabla_{\theta_i} \log \pi_{\theta_i}(a_i \mid o_i) Q_i^\pi(x, a_1, \cdots, a_N)
\end{aligned}
\tag{11}
$$

$Q_i^\pi(x, a_1, \cdots, a_N)$ is a centralized action–value function. The input consists of the state $x$, and the action $a_1, \cdots, a_N$ of all agents. The state $x$ can be simply composed of the observations of all agents. Each agent can design a reward function independently and learn independently to achieve competitive, cooperative, or hybrid policies.

Similar to DDPG, its policies are

$$
\boldsymbol{\mu} = \{\mu_1, \mu_2, \mu_3, \ldots, \mu_N\}
\tag{12}
$$

The parameters of the policies are

$$
\boldsymbol{\theta} = \{\theta_1, \theta_2, \theta_3, \cdots, \theta_N\}
\tag{13}
$$

The gradient of the objective function is

$$
\nabla_{\theta_i} J(\boldsymbol{\mu}_i) = \left[ \mathbb{E}_{x,a \sim D} \nabla_{\theta_i} \boldsymbol{\mu}_i(a_i \mid o_i) \nabla_{a_i} Q_i^\mu(x, a_1, \ldots, a_N) \Big|_{a_i = \boldsymbol{\mu}_i(o_i)} \right]
\tag{14}
$$

The experience replay buffer $D$ contains $(x, x', a_1, \cdots, a_N, r_1, \cdots, r_N)$, which records the experience of all agents. The loss function of the actor network is

$$
L(\boldsymbol{\mu}_i) = -Q_i^\mu(x, a_1, \cdots, a_N)
\tag{15}
$$

Accordingly, the critical network $Q_i^\mu$ is updated as follows:

$$
\begin{aligned}
L(\theta_i) &= \mathbb{E}_{x,a,r,x'} \left[ \left( Q_i^\mu(x, a_1, \ldots, a_N) - y \right)^2 \right] \\
y &= r_i + \gamma Q_i^{\mu'}(x', a_1', \ldots, a_N') \Big|_{a_j' = \mu_j'(o_j)}
\end{aligned}
\tag{16}
$$

where $\boldsymbol{\mu}' = \left\{ \mu_{\theta_1'}, \ldots, \mu_{\theta_N'} \right\}$ is the policy target network and $\theta_i'$ is the parameter of network $i$.

The MADDPG algorithm provides a common centralized training and distributed execution framework in multi-agent systems, as shown in Figure 3. However, the input dimension of the critical network will expand rapidly with the increase in the number of agents. Therefore, MADDPG cannot be applied to large-scale agent scenarios directly. Meanwhile, MADDPG may fail when the training and application scenarios are different. Based on the above discussion, this paper will solve the problems with the MADDPG algorithm and propose an extensible UAV swarm task assignment algorithm.

## 3. Extensible Task Assignment Algorithm of UAV Swarm

This section designs a scalable UAV swarm task assignment algorithm based on the following scenarios, which is trained on a small number of agents but can be directly applied to a larger UAV swarm system with guaranteed task assignment performance.

(1)   The UAV swarm searches for an unknown number of targets in a given area, using the Boids [36] algorithms to avoid obstacles during exploration.
(2)   The UAV is the ammunition to attack the detected target.
(3)   Each target needs multiple UAVs to destroy.

In this section, we design a local communication model and a mean simulation observation model to reduce the computational burden of the basic MADDPG algorithm. Meanwhile, we propose a parameter synchronization training mechanism, which guaran-

tees that the training network can be used in more UAVs directly. To ensure the performance, this paper proposes a multi-task assignment decision process. The system framework of the proposed Ex-MADDPG algorithm is shown in Figure 4, where letters A–F indicate the UAV swarm and the stars indicate the targets.

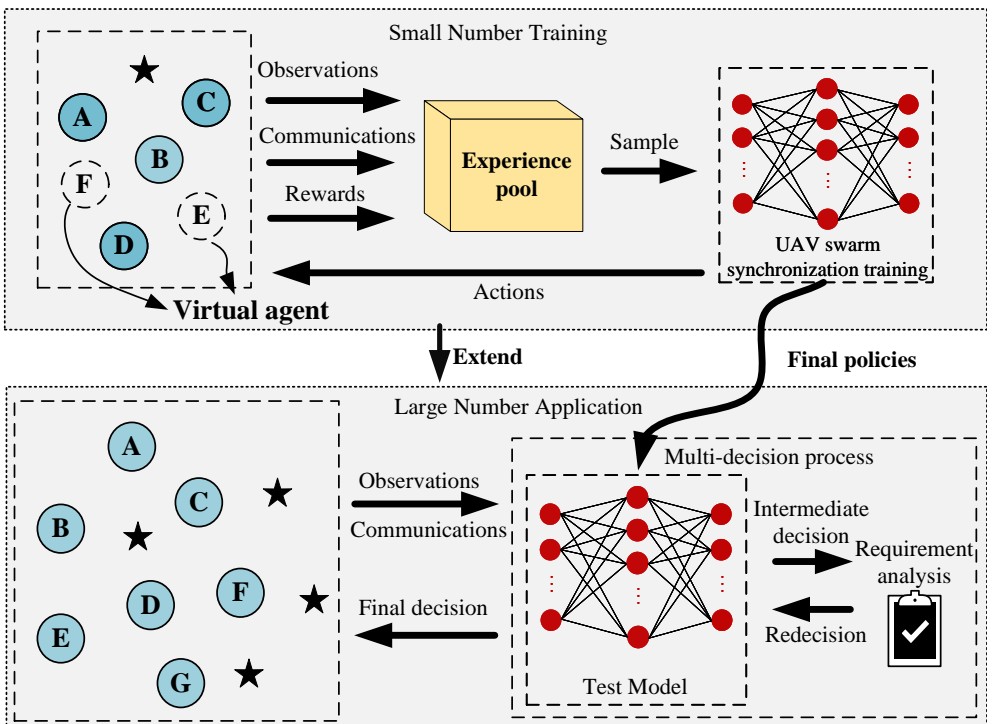

**Figure 4.** Extensible Task-Assignment Algorithm System Framework.

### 3.1. Local Communication Model

To ensure that the algorithm can be used in a large-scale UAV swarm system, this paper assumes that each agent can only receive partial information from its neighboring agents. The local communication model followed by each agent is designed as shown in Equation (17):

$$c_i^{out} = \left[ a_t, \ \boldsymbol{pos}_i, \ \boldsymbol{pos}_{i,goal} \right] \tag{17}$$

$$c_i^{in} = \begin{cases} \boldsymbol{c}_j^{out} & agent_j \ \text{in} \ C_{agent_i} \\ 0 & agent_j \ \text{not in} \ C_{agent_i} \end{cases} \tag{18}$$

where $c_i^{out}$ is the communication message sent from the agent $i$; $a_t$ is a bool variable, indicating whether the agent $i$ is in the ready attack state. $\boldsymbol{pos}_i$ is the position of the agent $i$; $\boldsymbol{pos}_{i,goal}$ is target positions found by the current agent $i$; $c_j^{out}$ is the release information of agent $j$; $c_i^{in}$ is the information received by the agent $i$; $C_{agent_i}$ refers to the communication range of agent $i$. An example of the local communication model is shown in Figure 5. The agent only receives messages within its communication range, such as $A$ and $B$. If two agents are not within each other's communication range, such as $B$ and $C$ in Figure 5, they will not communicate.

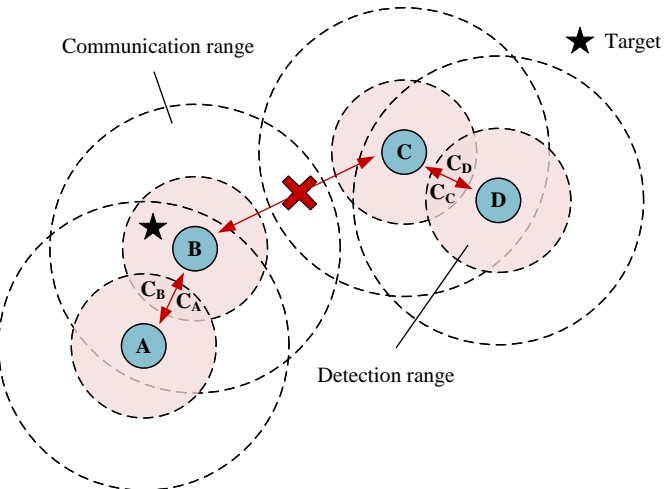

**Figure 5.** Schematic Diagram of Communication Mode.

### 3.2. Mean Simulation Observation Model

Aiming at the problem that the observation dimension changes with the scale of the UAV swarm, which leads to the failure of the DRL algorithm, this paper proposes fixed-dimension observation values to solve this problem. Compared with the Long Short-Term Memory (LSTM) method proposed by Zhou Wenqin [37], fixed dimension observation values are more stable in environments with huge changes. The design observation model is a mean simulation observation model, as shown in Equation (19):

$$obs_i = \left[ n,\ pos_i,\ pos_{mean},\ pos_{goal} \right] \tag{19}$$

The agent receives the target assignment $n$ according to local communication and dynamically adjusts its assignment strategy according to $n$. Its position $pos_i$, the average position of the surrounding agents $pos_{mean}$, and the target position $pos_{goal}$ allow the agent to complete the target assignment based on its own observations. Meanwhile, the dimensions of the observation will not change when the number of surrounding agents changes dynamically.

At the same time, $n$ is designed as a simulation quantity for large UAV swarms, where $n$ is simulated as a random number. Through the parameter n, the situation of a large number of UAVs around a single UAV can be simulated. By training a small number of UAVs, an algorithm suitable for a large number of scenarios can be obtained. The mean simulation observation model effectively reduces the computing power and time consumed by training a large number of UAV task assignment algorithms and solves the disadvantage that the trained algorithms can only be applied to a fixed number. In subsequent experiments, it can be proved that the algorithm using the mean simulation observation model greatly improves the scalability of the MADDPG algorithm.

### 3.3. Swarm Synchronization Training

In traditional multi-agent reinforcement learning training processes, the parameters of the agents are different, so the trained agents cannot be applied to systems of different scales. After the training is completed, the strategy of a single agent is often incomplete and needs the cooperation of other agent strategies. This training method can complete most multi-agent tasks. When considering the scalability, this training method will fail.

Inspired by the characteristics of bee colony systems, this paper designs a training mechanism called a swarm synchronization training mechanism, which is shown in Figure 6, to achieve scalability. Unlike the traditional reinforcement learning training process, all agent parameters are synchronized every certain number of training steps. Under the mean value

simulation observation model $\boldsymbol{obs}_i$, action value $Ac_i$, and the UAV swarm synchronization training mechanism, we obtain the gradient of the objective function:

$$\nabla_{\theta_i} J(\boldsymbol{\mu}_i) = \mathbb{E}_{\boldsymbol{x},Ac\sim D}\left[\nabla_{\theta_i}\boldsymbol{\mu}_i(Ac_i \mid \boldsymbol{obs}_i)\nabla_{Ac_i}Q_i^{\mu}(\boldsymbol{x},Ac_1,\ldots,Ac_N)\Big|_{Ac_i=\boldsymbol{\mu}_i(\boldsymbol{obs}_i)}\right] \tag{20}$$

where $\boldsymbol{x} = [obs_1,\ldots,obs_N]$ is the collection of the observations for each agent. The loss function of the actor network is rewritten in Equation (21):

$$L(\boldsymbol{\mu}_i) = -Q_i^{\mu}(\boldsymbol{x},Ac_1,\cdots,Ac_N) \tag{21}$$

The update method of critical network $Q_i^{\mu}$ is formulated as Equation (22):

$$L(\theta_i) = \mathbb{E}_{\boldsymbol{x},Ac,r,\boldsymbol{x}'}\left[\left(Q_i^{\mu}(\boldsymbol{x},Ac_1,\ldots,Ac_N) - \boldsymbol{y}\right)^2\right]$$
$$\boldsymbol{y} = \boldsymbol{r}_i + \gamma Q_i^{\mu'}\left(\boldsymbol{x}',Ac_1',\ldots,Ac_N'\right)\Big|_{Ac_j'=\boldsymbol{\mu}_j'(\boldsymbol{obs}_j)} \tag{22}$$

where $\boldsymbol{\mu}' = \left\{\boldsymbol{\mu}_{\theta_1'},\ldots,\boldsymbol{\mu}_{\theta_N'}\right\}$ is the actor target network, $\theta'$ is the actor target network parameter, $\omega$ is the critical network parameter, and $\omega'$ is the critic target network parameter. After a certain number of steps, we synchronize the parameters of the actor and critic network:

$$\boldsymbol{\theta}_{i\_new} = \frac{\sum_{j=1}^{N}\boldsymbol{\theta}_j}{N},\ \boldsymbol{\theta}_{i\_new}' = \frac{\sum_{j=1}^{N}\boldsymbol{\theta}_j'}{N}$$
$$\boldsymbol{\omega}_{i\_new} = \frac{\sum_{j=1}^{N}\boldsymbol{\omega}_j}{N},\ \boldsymbol{\omega}_{i\_new}' = \frac{\sum_{j=1}^{N}\boldsymbol{\omega}_j'}{N} \tag{23}$$

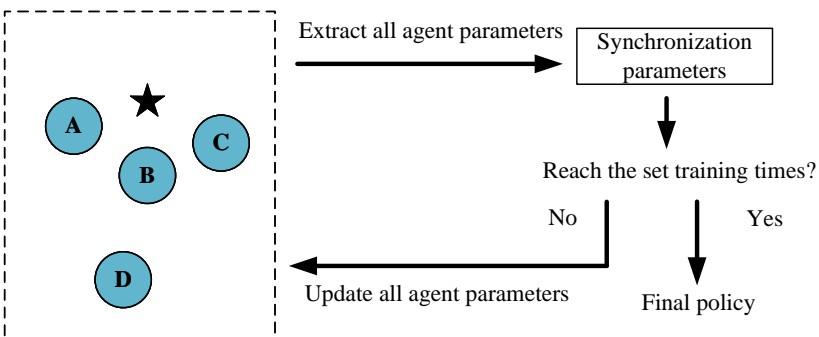

**Figure 6.** Swarm Synchronization Training Mechanism.

When the training process is finished, all agents have the same "brain", which means the same parameters. The final strategy $\boldsymbol{\mu}$ can be directly applied to any scale of a UAV swarm system. Thus, Ex-MADDPG with a synchronization training mechanism solves the problem of applying the algorithm to large-scale agents.

### 3.4. Extensible Multi-Decision Mechanism

To ensure the performance of dynamic task assignment, this paper proposes a multi-decision mechanism to adjust its decision in real time according to $n$, as shown in Figure 7. In this mechanism, all agents complete the first-round decision, i.e., $n = 0$, as shown in Equation (24), and then communicate with other involved agents (shown as the same color in Figure 7), and make an attack decision again, as shown in Equations (25) and (26):

$$[0, \boldsymbol{pos}_i, \boldsymbol{pos}_{mean}, \boldsymbol{pos}_{goal}] \rightarrow \boldsymbol{\mu}_i \rightarrow Ac_i \tag{24}$$

$$\sum_{i=1}^{N} Ac_i \rightarrow n \tag{25}$$

$$[n, \boldsymbol{pos}_i, \boldsymbol{pos}_{mean}, \boldsymbol{pos}_{goal}] \rightarrow \boldsymbol{\mu}_i \rightarrow Ac_i \tag{26}$$

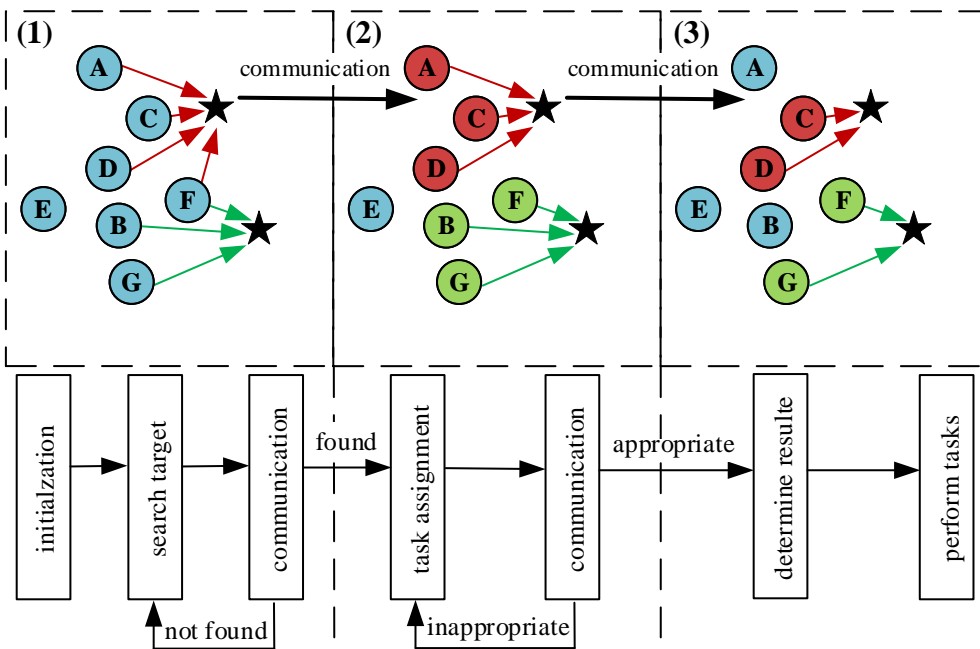

**Figure 7.** Scalable Multi-decision Mechanism.

**Remark 1.** *In practice, there may be more than one type of UAV to complete the mission and more than one target to attack. At the same time, the mission objective may have different priorities. It is impossible to train for every possible situation. The proposed multi-decision mechanism can easily accommodate these requirements by simply adding the appropriate decision conditions in Figure 7, such as the priority of the targets, the characteristics of attack targets, and so on.*

By designing multi-decision mechanisms, the algorithm is more scalable and can handle much more complex dynamic missions.

## 4. Simulation Experiments and Results

The simulation environment adopts the classical multi-agent simulation environment MPE, which is a set of 2D environments with discrete time and continuous space developed by OpenAI. This environment performs a series of tasks by controlling the motion of various role particles in 2D space [38]. At present, it is widely used in the simulation and verification of various multi-agent RL algorithms. We deployed the proposed algorithm on the PC platform with Intel Xeon Gold 5222 and NVIDIA GeForce RTX 2080Ti.

In this paper, the training scenario and the application scenario are made to be not exactly the same in order to illustrate the scalability of the proposed algorithm. Therefore, the training scenario and the application scenario will be discussed respectively.

### 4.1. Training Experiment Scenario

We trained the algorithm with only four agents and deploy the result to any large system. During the training process, the agent moves and searches for the target in the scene. After the agent finds the target, it will communicate with the surrounding agents and make a decision.

We set the following conditions and assumptions for the task-assignment training experiment:

(1) The UAV makes decisions based on the distance to the target and the average distance of the group to the target.
(2) The target needs at least two UAVs to destroy.
(3) The UAV only observes the target within its detection range.
(4) The UAV communicates only with agents within the communication range.

*4.2. Construction of Training Model*

During the training process, the training model is designed according to the task assignment conditions and specific simulation application scenarios in Section 4.1, including action, reward, mean simulation observation model, a swarm synchronization training mechanism, and a multi-decision mechanism.

4.2.1. Action Value

In the process of UAV swarm task assignment, the UAV needs to perform actions such as obstacle avoidance and assignment according to the observation value. According to the task requirements and experimental assumptions, the design action value is shown in Equation (27):

$$Ac = \left[ at, a_x, a_y \right] \tag{27}$$

where $a_x$ is the acceleration along the $x$ axis and $a_y$ is the acceleration along the $y$ axis. With $a_x$ and $a_y$, agents can move in any direction and at any speed on the plane.

4.2.2. Mean Simulation Observation

According to the task assignment conditions and subsequent expansion requirements, the analog quantity $n$ in the observation value in Section 3.2 is designed as shown in Equation (28):

$$n = \begin{cases} n & n < 3 \\ \text{random}(3, 32) & n \geq 3 \end{cases} \tag{28}$$

If the agent finds less than 3 agents attacking the same target through communication, the agent uses the accurate number. If the number is greater than or equal to 3, the agent uses a random number function to simulate it. In this way, the observation is still applicable in large systems.

4.2.3. Centralized and Distributed Reward Function

According to the scenario requirements of the UAV swarm task assignment, the design centralizes rewards for all agents:

$$R_{centralized} = \begin{cases} -2 & n = 0 \\ -\frac{D_{attack}}{D_{total}} - 0.03 \times 3 & n = 1 \\ -\frac{D_{attack}}{D_{total}} - 0.03 \times 2 + 1 & n = 2 \\ -\frac{D_{attack}}{D_{total}} - 0.03 \times n & \text{else} \end{cases} \tag{29}$$

where $D_{attack}$ means the average distance from the assigned agent to its target and $D_{total}$ is the average distance from all agents to the target. The larger the ratio, the worse the effect of assignment results. At the same time, based on the requirement that each target needs two UAV attacks, $R_{centralized}$ is divided into four parts according to the range of $n$ observed by the agent. When $n = 0$, it is at the beginning of the target assignment, and a large negative value is needed to promote the agent to make decisions. When $n = 2$, there will be a large positive reward to encourage such a decision Other cases will be punished according to the difference between 2. The greater the difference, the greater the punishment, and the agent will adjust its decision.

If the reward is too sparse under the above design, it will be difficult for the agent to learn the correct policy. Therefore, the distributed reward function $R_i$ of each agent is designed as in Equation (30) to guide the agent to move toward the target. In the process of

movement, the position and decision of the agent will change. In this process, the group will continue to optimize the decision and will improve the learning efficiency greatly.

$$R_i = -D_{i\_to\_goal} \tag{30}$$

where $D_{i\_to\_goal}$ represents the distance from the current agent to the target. The two reward functions are combined according to the following formula, and the total reward function $R$ is

$$R = R_i + 2 \times R_{centralized} \tag{31}$$

The training process is shown in Section 3.3 and Figure 6. The specific design is to synchronize the parameters of all agents every 10 training times. Meanwhile, the decision mechanism is introduced in Section 3.3 and Figure 7.

### 4.3. Validity of the Algorithm

The algorithm is mainly used in extended scenarios, i.e., scenarios where the number of agents and the number of targets are uncertain. In terms of the effectiveness of the algorithm, this paper will compare it with MADDPG, the mean simulated MADDPG (ms-MADDPG) algorithm, and the Hungarian algorithm.

The Hungarian algorithm is a combinatorial optimization algorithm that can solve the task-assignment problem in polynomial time. In the training scenario, the Hungarian algorithm is guaranteed to obtain an optimal assignment. Therefore, it is reasonable to choose the Hungarian algorithm for comparison.

According to the requirements of task-assignment scenarios, the task-assignment performance consists of task loss and task target difference. The task loss function is shown in Equation (32), and the task target difference is shown in Equation (33):

$$score_D = \frac{D_{attack}}{D_{all}} \tag{32}$$

$$score_A = |n - 2| \tag{33}$$

where $D_{attack}$ is the average distance from the attacking agent to the target and $D_{all}$ is the average distance from the participating agent to the target. In this scenario, since the agent needs to assign the target as quickly as possible, the distance ratio is used as the task loss instead of the absolute value of the distance. $n$ is the number of assignments to the current target. The greater the difference between the result of task assignment and the set value of 2, the greater the difference between the task target. Obviously, the smaller the task loss and task target difference, the better the performance of the task assignment algorithm.

As shown in Figure 8, Ex-MADDPG, MADDPG, ms-MADDPG have shown a significant decrease in task loss with the increase in training steps, and the algorithm can complete the task assignment according to the constraints. In Figure 9, the algorithm can be stable around 0.2 in the task-target difference. This shows that the algorithm can satisfy the task constraints of two agents attacking a target. The Hungarian algorithm can be designed as two agents attacking one target, and the difference is 0.

The results show that the Ex-MADDPG, ms-MADDPG, MADDPG, and Hungarian algorithms proposed in this paper can solve the task-assignment problem in the training scenario.

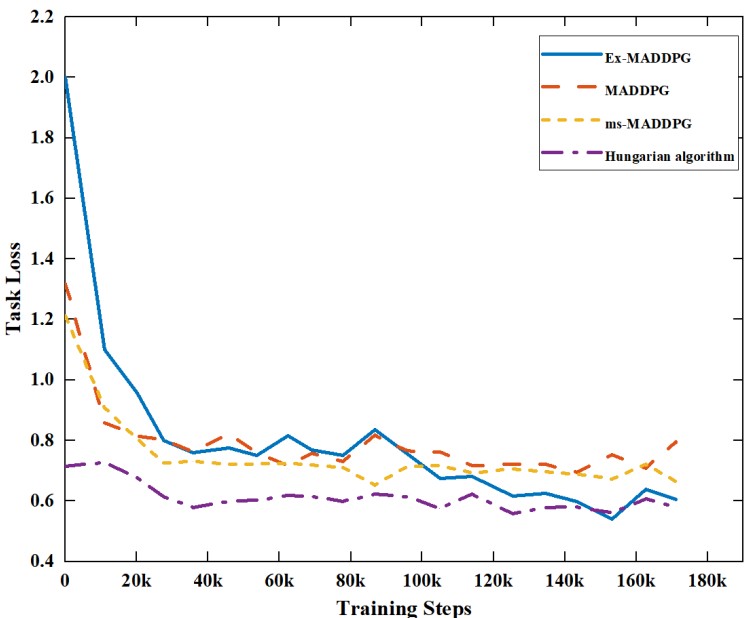

**Figure 8.** Task Loss.

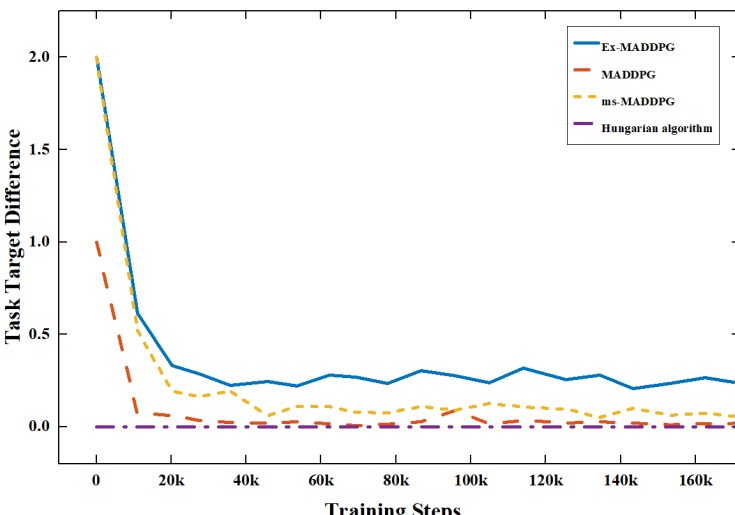

**Figure 9.** Task Target Difference.

*4.4. Extended Experiments*

Based on the training scenario in Section 4.1, we expand the number of agents and targets. At the same time, to increase the detection rate of the target, the expanded scene is divided into three phases: takeoff, search, and decision. The agent will take off from the bottom left, pass through the formation, fly to the target area and hover, and complete the total task.

In this section, we design two kinds of extension experiments to demonstrate the superiority of our proposed algorithm.

Experiment 1: Number extension

Considering that the attack ratio between agents and targets in the training process is 2:1, the number of agents and the number of targets are expanded according to this ratio. The experimental design conditions are as follows:

(1) Number of agents: number of targets = 2:1;
(2) The number of agents is between 8 and 56, with an interval of 4;

(3)   The final decision distance of the agent is designed as 1.5. If the distance between the agent and the target exceeds this range, the agent will communicate and make a decision at specified intervals. Otherwise, it will attack the target directly.

Experiment 2: Task extension

The practical application may be different from the training, so the tasks may also need to be expanded. To ensure the attack effect, we add task redistribution requirements, which means that one target may require two or more UAVs to destroy.

(1)   Number of agents:number of targets = 5:2;
(2)   The number of agents is between 5 and 55, with an interval of 5;
(3)   The final decision distance of the agent is chosen to be 1.5. If the distance between the agent and the target exceeds this range, the agent will communicate and make a decision at specified intervals. Otherwise, it will attack the target directly;
(4)   The target attacked by two agents has an 80% chance of being destroyed, while the target attacked by three agents has a 100% chance of being destroyed.

To reduce the random effect, each experiment is performed 30 times.

### 4.5. Extended Performance Test

Based on the performance requirements of the extended extension experiment, the extended performance metrics must be redesigned to compare the performance changes during the two extended experiments. In the following experiment, the 5:2 in the legend is used to refer to Experiment 2 for algorithm comparison and analysis.

The running simulation process of the algorithm is shown in Figure 10, which is applied to the dynamic target-assignment scenario of 32 agents. In Figure 10, (1) is the swarm of intelligent agents flying from the takeoff area to the target area, and (2) is the agent that first detects the target and starts to make decisions and allocate the target. Other agents continue to search. Finally, (3) is the final assignment result. Each target is hit by two agents.

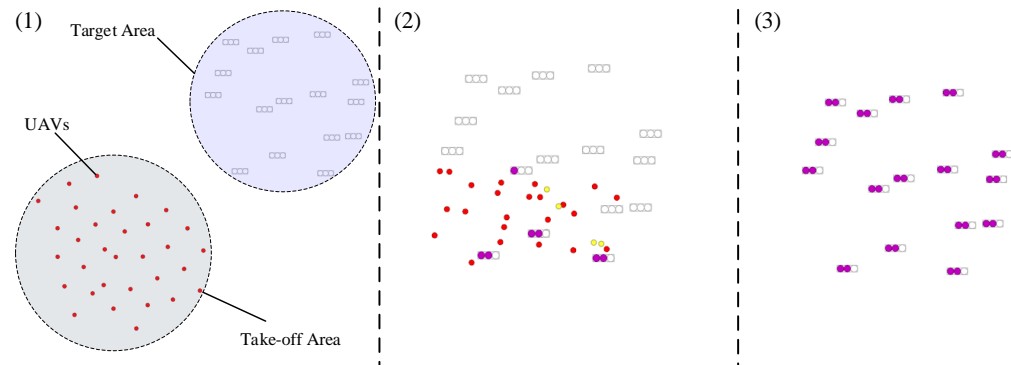

**Figure 10.** Algorithm Scalability Test Process.

For the above process, the following performance metrics are designed.

#### 4.5.1. Task Completion Rate

In the two extended experiments, the simplest way to judge the task completion is the target-destruction rate. In the process of task assignment, the more targets are destroyed, the higher the task completion rate will be. Therefore, the design task completion rate is given by Equation (34):

$$D_{rate} = \frac{N_{destroyed}}{N_{all}} \tag{34}$$

$N_{destroyed}$ refers to the number of completely destroyed targets, and $N_{all}$ refers to the number of all targets.

As shown in Figure 11, the improved MADDPG algorithm (ms-MADDPG) using only the mean simulation has a small scalability number. When the number of agents continues to increase, the task-completion rate of the ms-MADDPG algorithm decreases significantly, and it can only complete some tasks. The MADDPG algorithm has no scalability at all. When the number of applications is inconsistent with the number of training, the algorithm will not work, and the task completion rate is 0%.

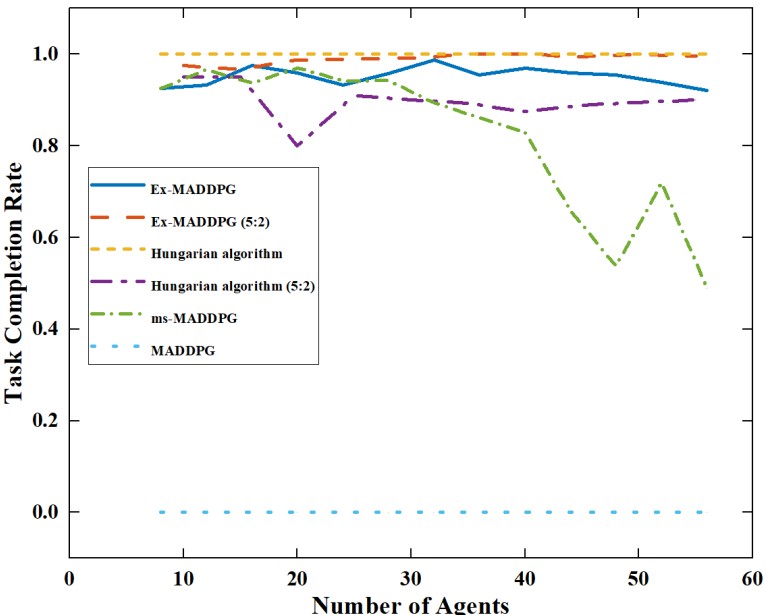

**Figure 11.** Task Completion Rate.

In the comparison test between Experiment 1 and Experiment 2 (5:2), the Hungarian algorithm was found to be able to destroy the target at 100% in Experiment 1, but destroyed less than 90% of the target in Experiment 2. At the same time, Ex-MADDPG could destroy more than 90% of the target in two experiments within the range of the number of tests. In Experiment 2, the algorithm could detect the target in real time. If the target was not completely destroyed, the new optimal agent was immediately determined to attack it to ensure that the target esd destroyed, achieving a task-completion rate of nearly 100%. The Ex-MADDPG algorithm could maintain the task assignment performance under application numbers and task-assignment conditions.

4.5.2. Task Loss

The design of task loss is the same as that of task loss in Section 4.3. This metric is the key metric for judging the distributional effect. The smaller the value, the more advantageous it is for the agent in the group, and the less time and distance it takes to execute the attack decision, which means the better the decision will be.

Figure 12 shows that the Hungarian algorithm, as a traditional algorithm, had poor results in Experiment 1 and Experiment 2. In the Hungarian algorithm, all agents make decisions at the same time, resulting in poor task loss performance. The Ex-MADDPG algorithm can select the agents in the group that is closer to the target to attack and make better decisions for each decision. When the shortest decision distance is chosen as 1.5, the agent can only make the final attack decision when it is relatively close to the target, which can further reduce the task loss. In terms of task loss, whether in Experiment 1 or Experiment 2, Ex-MADDPG algorithm has obvious advantages and can make better decisions in the case of expansion. However, the MADDPG algorithm cannot complete the expansion experiment and count its task loss.

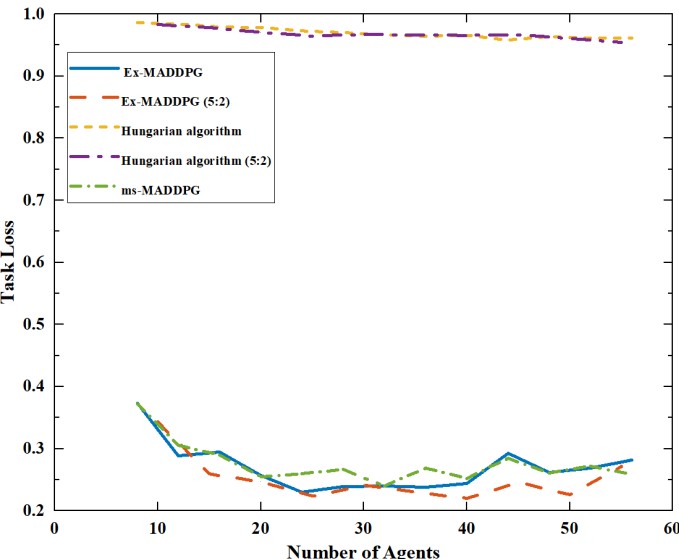

**Figure 12.** Task loss.

### 4.5.3. Decision Time

In the actual operation process, real time is a very important indicator that determines whether the agent can react quickly to external changes and react in real time. Therefore, the time required to execute a decision is designed as one of the performance metrics of the algorithm.

As shown in Figure 13, whether in Experiment 1 or Experiment 2, the Hungarian algorithm, as a traditional algorithm, has a relatively short execution time in a small number of cases, but there is an obvious upward trend with the increase in agents. It can be predicted that when the scale of the agent is large, it will take a lot of time to obtain the decision results. The Ex-MADDPG algorithm is calculated by a neural network. The change in the number of surrounding agents has little impact on its input value, and the number of iterations needed to make decisions is small. Therefore, with the increase in the number of agents, its decision-making time showed a small upward trend. It can easily meet the real-time requirements of the scene. However, the ms-MADDPG algorithm has many iterations to make decisions, and it is difficult for to make decisions, so it consumes the most time. The MADDPG algorithm cannot count its decision time because it cannot complete the expansion experiment.

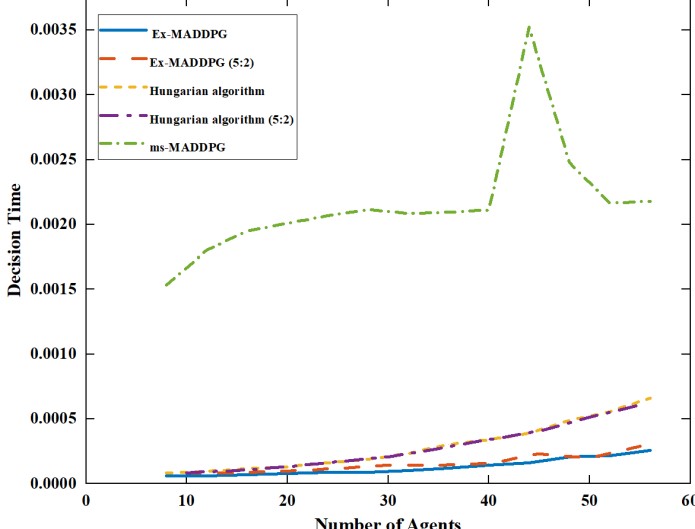

**Figure 13.** Decision Time.

### 4.5.4. Number of Communications

The number of agents that need to communicate refers to the number of other agents that each agent needs to communicate with when making decisions in a decision process. The more agents that need to communicate in a decision round, the more communication bandwidth is required to make decisions, the higher the hardware requirements for the agents, and the more difficult the algorithm is to implement.

As shown in Figure 14, in Experiment 1 and Experiment 2, the Hungarian algorithm needed all scene information to make decisions, and each agent needed to communicate with all other agents. Therefore, the number of communications is equal to the size of the agent. The Ex-MADDPG algorithm only needs to communicate with nearby agents and can make decisions using part of the scene information. The required communication bandwidth is therefore greatly reduced. The improved ms-MADDPG algorithm using mean simulation requires a medium number of agents to make decisions. It can be found that the introduction of neural networks greatly reduces the amount of information required for decision-making and reduces the communication bandwidth.

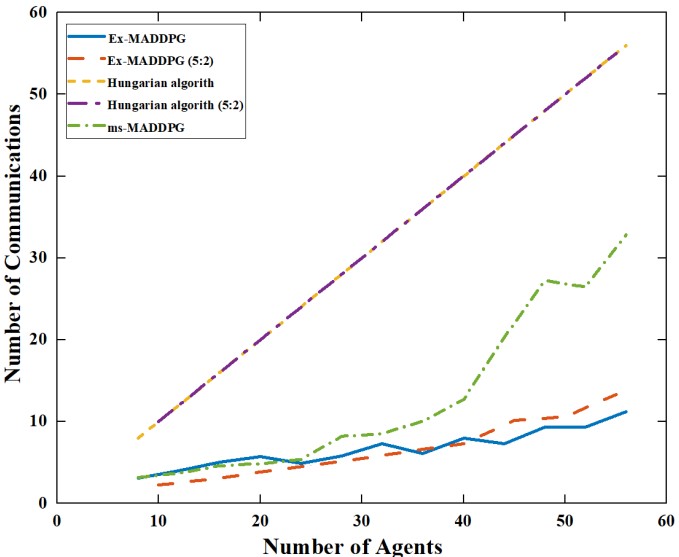

**Figure 14.** Number of Communications.

In conclusion, in Experiment 1 and Experiment 2 of the expansion experiment, the Ex-MADDPG algorithm was shown to be significantly superior to the traditional Hungarian algorithm and the MADDPG algorithm in terms of task completion rate, task loss, decision time, and the communication number, and could maintain stable performance during the expansion process and correctly complete the expected tasks.

## 5. Experiments and Results

In this section, experiments are transferred from simulation to the real world in the context of task assignment in a UAV swarm target-attacking scenario. To validate its performance in practical task assignment, experiments in the real world were conducted with a group of nano-quadcopters named scit-minis (as shown in Figure 15) flying under the supervision of a NOKOV motion capture system. We deployed the proposed algorithm on the same PC platform as the simulation, but the algorithm ran separately for each UAV. The scit-mini is a nano-quadcopter such as crazyfile 2.0 [39], but with much more power and a longer endurance. Meanwhile, we used an open-source, unmanned vehicle Autopilot Software Suite called ArduPilot to make it easier to transfer the algorithm from simulation to the real world.

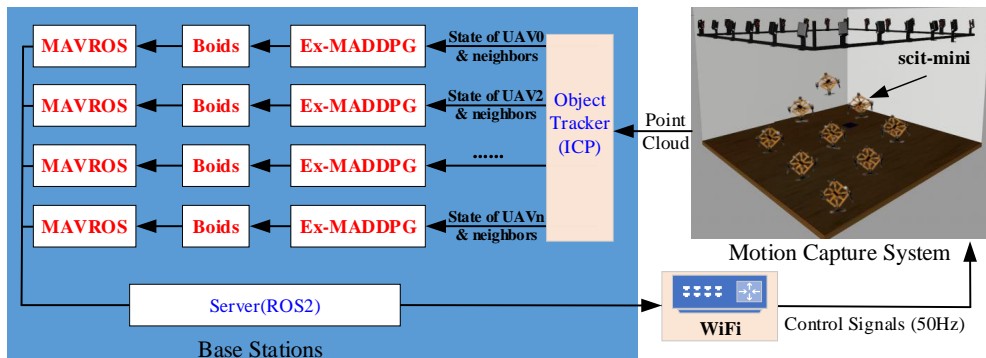

**Figure 15.** Diagram of system components.

### 5.1. Architecture Overview

Similar to crazyswarm [39], our system architecture is outlined in Figure 15. We tracked the scit-mini with a motion capture system using passive spherical markers. Thanks to the sub-millimeter recognition accuracy of the motion capture system, we used the Iterative Closest Point (ICP) algorithm [40] to obtain the precise location of each scit-mini in the swarm in real time.

Unlike crazyswarm, the scit-mini communicates with a PC platform over WiFi that can transfer more data than Crazyradio PA. The control signals run at 50 Hz with an ROS2 a communication delay of about 10~20 ms. However, the main onboard loop, like its attitude control loop, runs at 400 Hz, which can ensure the stable operation of the scit-mini. Each scit-mini obtains the control signals from its own Ex-MADDPG and boids through MAVROS with ROS2. We used only one computer in this experiment, but our system architecture supports multiple computers running simultaneously.

### 5.2. Flight Test

The actual test environment is shown in Figure 16, and the experimental area was divided into the take-off area and the target area. We tested our proposed Ex-MADDPG in a task-extension experiment, similar to Experiment 2 in Section 4.4, with nine scit-minis and three targets. The experimental procedure was as follows: nine scit-minis took off from the takeoff area shown in Figure 16(1) and then flew toward the target area, detected three targets, and executed the Ex-MADDPG algorithm. The experimental subject would fly over the target if it decides to attack it, and the rest of the scit-minis that do not decided to attack would continue to fly forward until they cross the target area. The scit-minis used the Boids algorithm to avoid collisions with each other.

We selected two key steps of the experiment during the whole autonomous process, as shown in Figure 16(2),(3). Figure 16(2) shows that the scit-mini made the first decision to attack its target using Ex-MADDPG, and Figure 16(3) is the final result of the task assignment. As shown in Figure 16(3), one target was attacked by two scit-minis and two targets were attacked by three scit-minis. The videos of this experiment and more tests are available online: https://youtu.be/shA1Tu7VujM.

The experiment demonstrated that the proposed Ex-MADDPG algorithm can accomplish task assignment in UAV swarm target-attacking scenario efficiently in real-time, verifying its practicality and effectiveness.

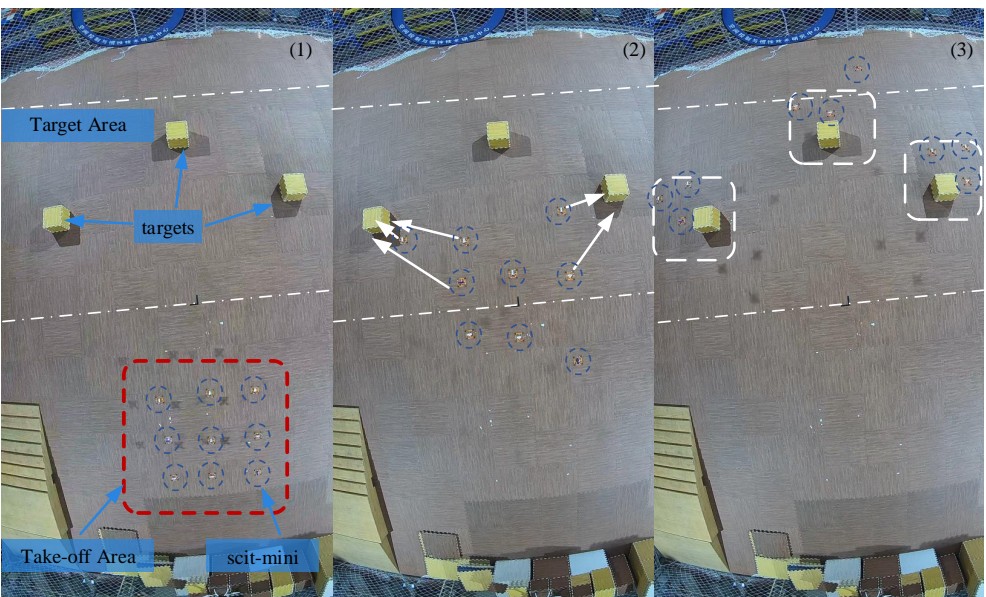

**Figure 16.** Flight Test for Ex-MADDPG.

## 6. Conclusions

This paper presents an improved algorithm Ex-MADDPG algorithm based on MAD-DPG to solve the problem of task assignment in a UAV swarm target-attacking scenario. This algorithm uses mean simulation observation and swarm-synchronization mechanisms to deploy in arbitrary-scale systems, training only a small number of agents. By designing the scalable multi-decision mechanism, this algorithm can maintain its performance in the process of expansion and achieve arbitrary expansion of the number of UAVs. At the same time, the algorithm can achieve task expansion and can complete similar tasks that differ from the training process. The Ex-MADDPG algorithm can be trained once and applied to a large number of task-assignment scenarios, effectively solving the problem of insufficient scalability of the traditional RL/DRL algorithm. Simulation results show that the Ex-MADDPG has obvious advantages over the Hungarian algorithm in terms of assignment performance, fault tolerance, and real-time capabilities. At the same time, the algorithm has good scalability and maintains performance under the condition of number and task expansion. Furthermore, the proposed method proves to be feasible and effective in UAV swarm target attack scenarios in both simulations and practical experiments.

In this paper, we propose a scalable reinforcement learning algorithm to address the task assignment problem in variable scenarios, with a particular focus on UAV formation planning. While the current implementation uses the Boids algorithm for formation flying, the UAV formation algorithm is not presented in detail. Therefore, future work will concentrate on the design and implementation of advanced formation planning algorithms to improve the efficiency of target detection and task assignment.

**Author Contributions:** Conceptualization, B.L. and S.W.; methodology, B.L. and S.W. and Y.P.; software, S.W. and Y.P.; validation, S.W., Y.P. and X.Z.; formal analysis, B.L.; investigation, B.L.; resources, Q.L.; data curation, C.W.; writing—original draft preparation, S.W. and B.L.; writing—review and editing, S.W., B.L. and Q.L.; visualization, S.W.; supervision, B.L. and C.W.; project administration, C.W. and Q.L.; funding acquisition, Q.L. and C.W. All authors have read and agreed to the published version of the manuscript.

**Funding:** This work was funded by the Touyan Innovation Program of Heilongjiang Province, China.

**Informed Consent Statement:** Informed consent was obtained from all subjects involved in the study.

**Data Availability Statement:** The data used to support the findings of this study are available from the corresponding author upon request.

**Conflicts of Interest:** The authors declare no conflict of interest.

**Abbreviations**

| | |
|---|---|
| MPE | Multi-Agent Particle Environment |
| UAV | Unmanned Aerial Vehicle |
| GA | Genetic Algorithm |
| SA | Simulated Annealing |
| ACO | Ant Colony Optimization algorithm |
| PSO | Particle Swarm Optimization algorithm |
| GW | Grey Wolf |
| ICP | Iterative Closest Point |
| DRL | Deep Reinforcement Learning |
| RL | Reinforcement Learning |
| DL | Deep Learning |
| DQN | Deep Q Network |
| MDPs | Markov Decision Processes |
| LSTM | Long Short-Term Memory |
| PG | Policy Gradient |
| DDPG | Deep Deterministic Policy Gradient |
| MADDPG | Multi-Agent Deep Deterministic Policy Gradient |
| ms-MADDPG | Mean Simulated MADDPG |
| Ex-MADDPG | Extensible Multi-Agent Deep Deterministic Policy Gradient |

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
