# Peer review of "Task Assignment of UAV Swarms Based on Deep Reinforcement Learning"

_drones, doi:10.3390/drones7050297_

Round 1

Reviewer 1 Report

This paper proposed a novel algorithm, Ex-MADDPG, to better solve the problem of task assignment in UAV swarm target-attacking scenarios. The proposed algorithm improves the scalability issues with other algorithms by considering local communication, mean simulation observation, synchronous parameter training mechanism, and multiple decision mechanism. Numerical investigations are completed in OpenAI’s Multi-Agent Particle Environment. Overall, it is a nice paper with proper literature survey and solid discussions. I recommend this paper for publication, but I still have some questions,

1.      Page 1, line 29-31. It seems that "optimization algorithm" and "heuristic algorithm" overlap. Authors should provide the reference of this sentence.

2.      Page 9, line 264, “corresponding judgment conditions”. Can you elaborate the conditions?

3.      Page 9, Figure 7. It looks like that this figure wants to express a two-step strategy, ‘divide-and-conquer’. Is it right? The name, “multi-decision”, may not be suitable for this strategy.

4.      Page 12, line 359. Missing punctuation, ‘.’.

5.      Page 12, line 363-364, “Ex-MADDPG gains better extensibility by sacrificing a part of its training performance”. What is the main drawback of the proposed algorithm? Is it possible to tackle this issue in the future?

6.      Page 13, line 399-402. Is the trajectory considered in this paper? From Figure 10, it seems that the UAVs are sequentially placed around the targets, not simultaneously. Is it related to priority?

Author Response

Dear Reviewer,

Thank you very much for your time involved in reviewing the manuscript and your very encouraging comments on the merits.

Please see the attachment for detail reply.

Reviewer 2 Report

The problem being addressed in the paper proposing the "Extensible Multi-Agent Deep Deterministic Policy Gradient (Ex-MADDPG) algorithm" is how to effectively solve the task assignment problem for Unmanned Aerial Vehicle (UAV) swarms using deep reinforcement learning. Specifically, the authors aim to address the challenge of coordinating multiple agents (i.e., UAVs) to perform a task cooperatively and efficiently. The authors propose an extension to the MADDPG (Multi-Agent Deep Deterministic Policy Gradient) algorithm, which can handle varying swarm sizes and different task types, making it an extensible and versatile solution to the task assignment problem for UAV swarms. The Ex-MADDPG algorithm can optimize the task assignment policy of the swarm, allowing for effective coordination and task completion.

This approach deals with the centralized structure and lacks justification for a decentralized structure swarm using this Multi-Agent Deep Deterministic Policy Gradient approach. 

This algorithm is feasible only in simulation mode, lack of clarity in the paper for a real-world scenario.

How do authors address the scalability problem in this proposed solution? I did not find proper justification in this paper.

The quality of the paper is not up to mark and required improvement in the results part.

This paper presents significantly less research content. Only simulated results are not ok; the algorithm should align with the real-world scenario.

More results with various scenarios should be presented in the paper.

Authors shold add a list of abreviations used in the manuscript at the end of the paper.

Most of figures (8, 9, 11, 12, 13, 14) looks clumsy and unprofessional. Authors should use appropriate colour schemes, font size and keep in mind the proper space requirements of different components.

Author Response

Dear Reviewer,

We express our gratitude for your time spent reviewing the manuscript and for your encouraging comments on its strengths.

Please see the attachment for detail reply. 

Sincerely,

The Authors

Reviewer 3 Report

Well written article with interesting results. It is need recheck English (for instance, str.95 misprint a word "algorithem", str. 123, 144 should be started with small "w", str.248 "All" should be started with small "a")

Author Response

Dear Reviewer,

We appreciate the time you invested in reviewing the manuscript and your encouraging remarks on its strengths. Thank you very much.

We highly value your proposal as it plays a crucial role in enhancing the quality of our paper. Your feedback was clear and detailed, and we sincerely appreciate it. In the following sections of this letter, we will provide a detailed response to each of your comments.

Comment 1:

Well written article with interesting results. It is need recheck English (for instance, str.95 misprint a word "algorithem", str. 123, 144 should be started with small "w", str.248 "All" should be started with small "a")

Response 1:

Thank you for the detailed review. We have carefully and thoroughly proofread the manuscript to correct all the grammar and typos. And we have rechecked the entire article and had it revised by a professional agency. Thank you very much for pointing out these problems for us.

Please see the attachment for detail.

Sincerely,

The Authors

Reviewer 4 Report

-The paper although interesting, is related to a topic that has already been thoroughly studied and investigated. The standardization of this class of protocols is already very advanced and at least some indications should be provided in the text before acceptance.

-The novelty and major contribution of the research need properly described. 

-Another point that is very lightly treated but that should require more input is on the issues and cost of implementing these types of solutions.

-There is no related work section, which is just summarized in the introduction. This could be ok for a short conference paper, but definitely not for a full-length journal version with no page limitation. The authors should provide a full section explaining the novelty of their approach. Therefore, I leave some suggestions for related work: - Fine-tuning of UAV control rules for spraying pesticides on crop fields;  -Exploiting the use of unmanned aerial vehicles to provide resilience in wireless sensor networks

-A table could be made to make evident the contribution of this manuscript.

-I hope to read something about the practical application of the approach. A Section of applicability of the proposed solution is requerid in this manuscript.

Author Response

Dear Reviewer,

We express our gratitude for your time spent reviewing the manuscript and for your encouraging comments on its strengths.

Your proposal holds great significance for us in terms of enhancing the quality of the paper. In the remaining sections of this letter, we will delve into each of your comments individually and provide our corresponding responses.

Please see the attachment for detail reply.

Sincerely,

The Authors

Round 2

Reviewer 4 Report

The authors responded appropriately to the comments